# PanoOcc: Unified Occupancy Representation for Camera-based 3D Panoptic Segmentation

## Abstract

Comprehensive modeling of the surrounding 3D world is crucial for the success of autonomous driving. However, existing perception tasks like object detection, road structure segmentation, depth & elevation estimation, and open-set object localization each only focus on a small facet of the holistic 3D scene understanding task. This divide-and-conquer strategy simplifies the algorithm development process but comes at the cost of losing an end-to-end unified solution to the problem. In this work, we address this limitation by studying **camera-based 3D panoptic segmentation**, aiming to achieve a unified occupancy representation for camera-only 3D scene understanding. To achieve this, we introduce a novel method called **PanoOcc**, which utilizes voxel queries to aggregate spatiotemporal information from multi-frame and multi-view images in a coarse-to-fine scheme, integrating feature learning and scene representation into a unified occupancy representation. We have conducted extensive ablation studies to validate the effectiveness and efficiency of the proposed method. Our approach achieves new state-of-the-art results for camera-based semantic segmentation and panoptic segmentation on the nuScenes dataset. Furthermore, our method can be easily extended to dense occupancy prediction and has demonstrated promising performance on the Occ3D benchmark. The code will be made available.

## 1 Introduction

Holistic 3D scene understanding is vital in autonomous driving. The capability to perceive the environment, identify and categorize objects, and contextualize their positions in the 3D space of the scene is fundamental for developing a safe and reliable autonomous driving system.

Recent advancements in camera-based Bird's Eye View (BEV) methods have shown great potential in enhancing 3D scene understanding. By integrating multi-view observations into a unified BEV space, these methods have achieved remarkable success in tasks such as 3D object detection (Wang et al., 2022b; Li et al., 2022d; Liu et al., 2022c; Li et al., 2022c), BEV semantic segmentation (Philion & Fidler, 2020; Hu et al., 2021; Zhou & Krähenbühl, 2022), and vector map construction (Liu et al., 2022b; Liao et al., 2022a). However, existing perception tasks have certain limitations as they primarily focus on specific aspects of the scene. Object detection is primarily concerned with identifying foreground objects, BEV semantic segmentation only predicts the semantic map on the BEV plane, and vector map construction emphasizes the static road structure of the scene. To address these limitations, there is a need for a more comprehensive and integrated paradigm for 3D scene understanding. In this paper, we propose *camera-based panoptic segmentation*, which aims to encompass all the elements within the scene in a unified representation for the 3D output space.

However, directly utilizing Bird's Eye View (BEV) features for camera-based panoptic segmentation leads to poor performance due to the omission of finer geometry details, such as shape and height information, which are crucial for decoding fine-grained 3D structures. This limitation motivates us to explore a more effective 3D feature representation. Occupancy representation has gained popularity as it effectively describes various elements in the scene, including open-set objects (e.g., debris), irregular-shaped objects (e.g., articulated trailers, vehicles with protruding structures), and special road structures (e.g., construction zones). Therefore, a burst of recent methods (Cao & de Charette, 2022; Huang et al., 2023; Miao et al., 2023; Cao & de Charette, 2022; Wang et al., 2023a; Li et al., 2023) have focused on dense semantic occupancy prediction. However, simply lifting 2D to 3D

occupancy representation has been considered inefficient in terms of memory cost. This limitation has driven methods like TPVFormer (Huang et al., 2023) to split the 3D representation into three 2D planes. Although these methods attempt to mitigate the memory issue, they still struggle to capture the complete 3D information and may experience reduced performance. Moreover, these existing works primarily concentrate on the semantic comprehension of the scene and do not tackle instance-level discrimination. Fine-grained foreground segmentation is crucial for 3D perception.

In this work, we propose a novel method called *PanoOcc*, which seamlessly integrates object detection and semantic segmentation in a joint-learning framework, facilitating a more comprehensive comprehension of the 3D environment. Both detection and segmentation performance can benefit from this joint-learning framework. Our approach employs voxel queries to learn a unified occupancy representation. This occupancy is learned in a coarse-to-fine scheme, solving the problem of memory cost and significantly enhancing efficiency. We then take a step further to explore the sparse nature of 3D space and propose an occupancy sparsify module. This module progressively prunes occupancy to a spatially sparse representation during the coarse-to-fine upsampling, greatly boosting memory efficiency. Our contributions are summarized as follows:

- We introduce *camera-based 3D panoptic segmentation* as a new paradigm for holistic 3D scene understanding, which utilizes multi-view images to create a unified occupancy representation for the 3D scene. This allows us to jointly model object detection and semantic segmentation within a single end-to-end model, leading to a more cohesive and holistic understanding of the scene.

- Our proposed framework, PanoOcc, adopts a *coarse-to-fine scheme* to learn the unified occupancy representation from multi-frame and multi-view images. We demonstrate that using 3D voxel queries with a coarse-to-fine learning scheme is effective and efficient. This scheme could be further made spatially sparse to boost memory efficiency by an occupancy sparsify module.

- Experiments on the nuScenes dataset show that our approach achieves state-of-the-art performance on camera-based semantic segmentation and panoptic segmentation. Furthermore, our approach can extend to dense occupancy prediction and has shown promising performance on the Occ3D benchmark.

## 2 RELATED WORK

**Camera-based 3D Perception.** Camera-based 3D perception has received extensive attention in the autonomous driving community due to its cost-effectiveness and rich visual attributes. Previous methods perform 3D object detection and map segmentation tasks independently. Recent BEV-based methods unify these tasks on the problem of feature view transformation from image space to BEV space. One line of works follows the lifting paradigm proposed in LSS (Philion & Fidler, 2020); they explicitly predict a depth map and lift multi-view image features onto the BEV plane (Huang et al., 2021; Li et al., 2022c;b; Park et al., 2022). Another line of works inherits the spirit of querying from 3D to 2D in DETR3D (Wang et al., 2022b); they employ learnable queries to extract information from image features by cross-attention mechanism (Li et al., 2022d; Lu et al., 2022; Jiang et al., 2023; Wang et al., 2023b). While these methods efficiently compress information onto the BEV plane, they may sacrifice some of the integral scene structure inherent in 3D space. To address this limitation, voxel representation is better suited for obtaining a holistic understanding of 3D space, making it ideal for tasks such as 3D semantic segmentation and panoptic segmentation.

**3D Occupancy Prediction.** Occupancy prediction can be traced back to Occupancy Grid Mapping (OGM) (Thrun, 2002), a classic task in mobile robot navigation that aims to generate probabilistic maps from sequential noisy range measurements. Recently, there has been considerable attention given to camera-based 3D occupancy prediction, which aims to reconstruct the 3D scene structure from images. Existing tasks in this area can be categorized into two lines based on the type of supervision: sparse prediction and dense prediction. Sparse prediction methods obtain supervision from LiDAR points and are evaluated on LiDAR benchmarks. Huang et al. (2023) proposes a tri-perspective view method for predicting 3D occupancy. Dense prediction methods are closely related to Semantic Scene Completion (SSC) (Armeni et al., 2017; Song et al., 2017; Dai et al., 2017a; Liao et al., 2022b). MonoScene (Cao & de Charette, 2022) first uses U-Net to infer dense 3D occupancy with semantic labels from a single monocular RGB image. VoxFormer (Li et al.,

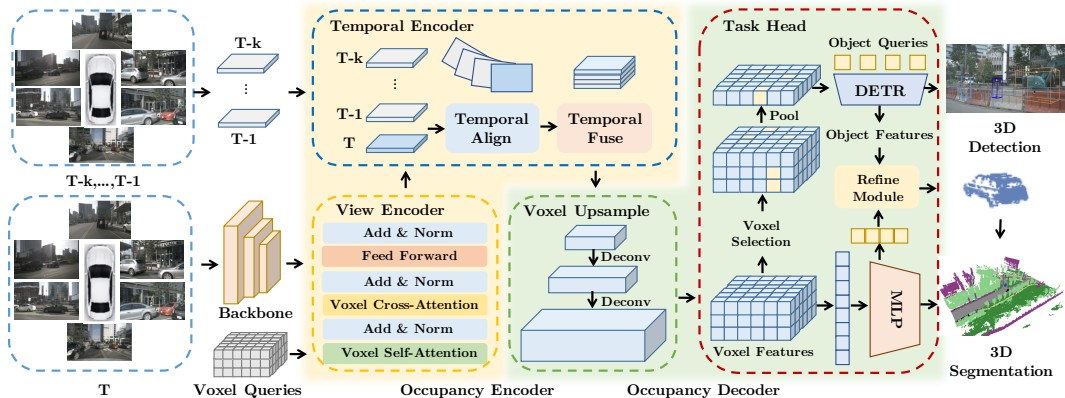

Figure 1: **The overall framework of PanoOcc.** Our framework begins by utilizing an image back-bone network to extract multi-scale features from multi-view images across multiple frames. Subsequently, voxel queries are employed to learn voxel features through the *View Encoder*. The *Temporal Encoder* then aligns the previous voxel features with the current frame and combines these features. *Voxel Upsample* restores the high-resolution voxel representation for fine-grained semantic classification. *Task Head* predicts object detection and semantic segmentation by two separate heads. *Refine Module* further refines the thing class prediction with the help of 3D object detection and assigns the instance ID to generate 3D panoptic segmentation results.

2023) utilizes depth estimation to select voxel queries in a two-stage framework. Subsequently, a series of studies have focused on the task of dense occupancy prediction and have introduced new benchmarks. OpenOccupancy (Wang et al., 2023a) provides a carefully annotated occupancy benchmark, while Occ3D (Tian et al., 2023) proposes an occupancy prediction benchmark using the Waymo and nuScenes datasets. Openocc (Tong et al., 2023) further provides occupancy flow annotation for dynamic objects modeling on the nuScenes dataset.

**LiDAR Panoptic Segmentation.** LiDAR panoptic segmentation (Milioto et al., 2020) offers a comprehensive understanding of the environment by unifying semantic segmentation and object detection. However, traditional object detection methods often lose height information, making it challenging to learn fine-grained feature representations for accurate 3D segmentation. Recent LiDAR panoptic methods (Zhou et al., 2021; Razani et al., 2021; Hong et al., 2021) have been developed based on well-designed semantic segmentation networks (Zhang et al., 2020; Cheng et al., 2021) to address this limitation. Instead of predicting sparse semantic segmentation on LiDAR points, camera-based panoptic segmentation aims to output dense voxel segmentation of the scene.

## 3 METHODOLOGY

### 3.1 PROBLEM SETUP

**Camera-based 3D panoptic segmentation** aims to predict a dense panoptic voxel volume surrounding the ego-vehicle using multi-view images as input. Specifically, we take current multi-view images denoted as $\mathbf{I}_t = \{\mathbf{I}_t^1, \mathbf{I}_t^2, ..., \mathbf{I}_t^n\}$ and previous frames $\mathbf{I}_{t-1}, ..., \mathbf{I}_{t-k}$ as input. $n$ denotes the camera view index, while $k$ denotes the number of history frames. The model outputs the current frame semantic voxel volume $\mathbf{Y}_t \in \{w_0, w_1, ..., w_C\}^{H \times W \times Z}$ and its corresponding instance ID $\mathbf{N}_t \in \{v_0, v_1, v_2, ..., v_P\}^{H \times W \times Z}$. Here $C$ denotes the total number of semantic classes in the scene, while $w_0$ represents the empty voxel grid. $P$ are the total number of instances in the current frame $t$; for each grid belonging to the foreground classes (*thing*), it would assign a specific instance ID $v_j$. $v_0$ is assigned to all voxel grids belonging to the *stuff* categories and empty. $H, W, Z$ denotes the length, width, and height of the voxel volume.

**Camera-based 3D semantic occupancy prediction** can be considered a sub-problem of camera-based 3D panoptic segmentation. The former focus only on predicting the semantic voxel volume $\mathbf{Y}_t \in \{w_0, w_1, ..., w_C\}^{H \times W \times Z}$. The emphasis is placed on accurately distinguishing the empty class ($w_0$) from the other classes to determine whether a voxel grid is empty or occupied.

### 3.2 OVERALL ARCHITECTURE

In this section, we introduce the overall architecture of PanoOcc, serving as a baseline for 3D panoptic segmentation. As illustrated in Figure 1, our approach takes multi-frame multi-view images as input and outputs 3D panoptic segmentation for the current scene. Firstly, the image backbone extracts multi-scale features of input images. These features are then processed by the *Occupancy Encoder*, which consists of the *View Encoder* and *Temporal Encoder*, to generate a coarse unified occupancy representation. Specifically, the *View Encoder* utilizes voxel queries to learn voxel features, preserving the actual 3D structure of the scene by explicitly encoding height information. The *Temporal Encoder* aligns and fuses previous voxel features with the current frame, capturing temporal information and enhancing the representation. The *Occupancy Decoder* employs a coarse-to-fine scheme to recover fine-grained occupancy representation. The *Coarse-to-fine Upsampling* module restores the high-resolution voxel representation, enabling efficient learning of precise occupancy representation. With the advantage of a unified occupancy representation, the model can jointly learn object detection and semantic segmentation through the *Task Head*. Finally, the *Refine Module* refine the prediction of *thing* classes and output 3D panoptic segmentation results.

Our model follows two key design principles: 1. **Unified occupancy representation** for learning and task output. 2. **Efficient feature learning** for 3D scene. In the following, we provide detailed descriptions of designs in these two aspects.

### 3.3 UNIFIED OCCUPANCY REPRESENTATION

Occupancy serves as a unified 3D representation, not only reflected in different tasks unity (object detection and semantic segmentation), but also in unifying feature learning and output spaces. Therefore we introduce our method from *Unified Learning* and *Unified Task* in the following.

**Unified Learning.** We adopt occupancy as feature representation in the learning process. To achieve this, we use *voxel queries* to aggregate multi-frame multi-view image features within *occupancy encoder*. Occupancy encoder consists of view encoder and temporal encoder. We define a group of 3D-grid-shape learnable parameters $\mathbf{Q} \in \mathbb{R}^{H \times W \times Z \times D}$ as voxel queries. $H$ and $W$ are the spatial shape of the BEV plane, while $Z$ represents the height dimension, and $D$ is the embedding dimension. A single voxel query $\mathbf{q} \in \mathbb{R}^D$ located at $(i, j, k)$ position of $\mathbf{Q}$ is responsible for the corresponding 3D voxel grid cell region. Each grid cell in the voxel corresponds to a real-world size of $(s_h, s_w, s_z)$ meters. Given voxel queries $\mathbf{Q}$ and extracted image feats $\mathbf{F}$ as input, the occupancy encoder outputs the fused voxel features $\mathbf{Q}_f \in \mathbb{R}^{H \times W \times Z \times D}$.

Compared to previous feature transformation based on BEV queries, the primary difference lies in the *attention operations* (Zhu et al., 2020) and *temporal alignments*. In *view encoder*, we adapt the attention operations to voxel space by designing voxel self-attention and voxel cross-attention. To lift the BEV queries to voxel queries computation, the core difference lies in the choice of *reference points*, details refer to A.1 in the appendix. *Temporal encoder* consists of two specific operations: *temporal align* and *temporal fuse*. Different from previous temporal alignment methods (Li et al., 2022d; Park et al., 2022), which align history features on the BEV plane, our approach employs voxel alignment in 3D space. This allows us to correct for the inaccuracies caused by the assumptions made in previous BEV-based methods that road height remains unchanged throughout the scene, which is not always valid in real-world driving scenarios, particularly when encountering uphill and downhill terrain. Voxel alignment is crucial for fine-grained voxel representations to perceive the environment accurately. Specifically, the process of voxel alignment is formulated as follows:

$$\mathbf{Q}_{t-k \to t} = \text{GridSample}(\mathbf{Q}_{t-k}, \mathbf{G}_{t-k}), \quad \mathbf{G}_{t-k} = \mathbf{T}_{t \to t-k} \cdot \mathbf{G}_t \tag{1}$$

where $\mathbf{G}_t \in \mathbb{R}^{H \times W \times Z}$ is the voxel grid at current frame $t$, $\mathbf{G}_{t-k} \in \mathbb{R}^{H \times W \times Z}$ represents the current frame grid at frame $t - k$. $\mathbf{T}_{t \to t-k}$ is the transformation matrix for transforming the points at frame $t$ to previous frame $t - k$. Then the queries at frame $t - k$ are aligned to current frame $t$ by interpolation sampling, denoted as $\mathbf{Q}_{t-k \to t}$. After the alignment, the previous aligned voxel queries $[\mathbf{Q}_{t-k \to t}, ..., \mathbf{Q}_{t-1 \to t}]$ are concated with the current voxel queries $\mathbf{Q}_t$. We employ a block of residual 3D convolution to fuse the queries and output fused voxel queries $\mathbf{Q}_f$.

**Unified Task.** With the advantage of occupancy representation, the model has a strong capacity to handle different tasks. We can unify the 3D object detection and semantic segmentation into 3D

panoptic segmentation, achieving a more comprehensive understanding of the scene and a finer-grained modeling of objects. This allow us to *train jointly* and benefit from each other through the *foreground information propagation*.

Specifically, our model is trained end-to-end for joint detection and segmentation while previous methods usually train separately due to the conflicting learning objectives. To address this problem, we leverage foreground occupancy to communicate between semantic head and detection head. The total loss $\mathcal{L}$ has two parts: $\mathcal{L}_{Det}$ and $\mathcal{L}_{Seg}$. The semantic voxel segmentation head is supervised by $\mathcal{L}_{Seg}$, a dense loss consisting of focal loss (Lin et al., 2017b) (all voxels) and Lovasz loss (Berman et al., 2018) (non-empty voxels). We adopt *voxel selection* to convey the foreground information to detection head, which predicts a binary voxel mask to select the foreground categories (*thing*) voxel features. The voxel mask is supervised by focal loss (Lin et al., 2017b) $\mathcal{L}_{thing}$. The total loss $\mathcal{L}_{Seg}$ is formulated as:

$$\mathcal{L}_{Seg} = \lambda_1 \mathcal{L}_{focal} + \lambda_2 \mathcal{L}_{lovasz} + \lambda_3 \mathcal{L}_{thing} \tag{2}$$

The detection head is supervised by $\mathcal{L}_{Det}$, a sparse loss consisting of focal loss (Lin et al., 2017b) for classification and L1 loss for bounding box regression:

$$\mathcal{L}_{Det} = \lambda_4 \mathcal{L}_{cls} + \lambda_5 \mathcal{L}_{reg} \tag{3}$$

*Refine module* further refines the predicted foreground (*thing*) voxels using the detection results and generate 3D panoptic segmentation results. We start by sorting all box predictions based on their confidence scores. Then, we select a set of high-confidence bounding boxes denoted as $G = \{b_i | s_i > \tau\}$, where $b_i$ represents a 3D bounding box, $s_i$ is the confidence score, and $\tau$ is a threshold (default: $\tau = 0.8$). For the voxels within each bounding box $b_i$, we assign the class prediction $c_i$ to all of them. To perform panoptic voxel segmentation, we assign instance IDs sequentially based on confidence scores. If the current instance overlaps with previous instances beyond a certain threshold, we ignore it to avoid duplication. Finally, we assign instance ID 0 to all voxels corresponding to the *stuff* class.

### 3.4 EFFICIENT FEATURE LEARNING

Compared the information density in image space, 3D space obvious exhibits greater sparsity. Besides, directly extending BEV features to voxel features would incur significant memory and computational costs. Therefore, we make two designs within *occupancy decoder*: *Coarse-to-fine Upsampling* and *Occupancy Sparsify* to mitigate this problem.

**Coarse-to-fine Upsampling.** This design enables us to only learn a coarse voxel feature $\mathbf{Q}_f$ in the occupancy encoder. This module upsamples the fused voxel query $\mathbf{Q}_f \in \mathbb{R}^{H \times W \times Z \times D}$ to the high-resolution occupancy features $\mathbf{O} \in \mathbb{R}^{H' \times W' \times Z' \times D'}$ by 3D deconvolutions. Such a coarse-to-fine manner not only avoids directly applying expensive 3D convolutions to high-resolution occupancy features, but also leads to no performance loss. We have a quantitative discussion in the Table 6.

**Occupancy Sparsify.** Although the coarse-to-fine manner guarantees the high efficiency of our method, there is a considerable computational waste on the spatially dense feature $\mathbf{Q}_f$ and $\mathbf{O}$. This is because our physical world is essentially sparse in spatial dimensions, which means a large portion of space is not occupied. Dense operations (i.e., dense convolution) violate such essential sparsity. Inspired by the success of sparse architecture in LiDAR-based perception (Yan et al., 2018; Liu et al., 2022a; Fan et al., 2023), we optionally turn to the Sparse Convolution (Graham & van der Maaten, 2017) for occupancy sparsify. In particular, we first learn an occupancy mask for $\mathbf{Q}_f$ to indicate if positions on $\mathbf{Q}_f$ are occupied. Then we prune $\mathbf{Q}_f$ to a sparse feature $\mathbf{Q}_{sparse} \in \mathbb{R}^{N \times D}$ by discarding those empty positions according to

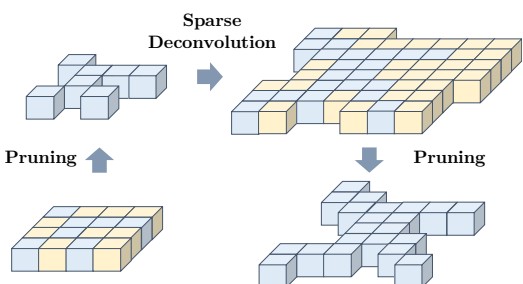

Figure 2: **Illustration of occupancy sparsify.** It serves as an optional technique to boost efficiency. We use BEV representation for simple illustration, while it is actually a 3D process. The light yellow region will be pruned according to occupancy masks.

the learned occupancy mask, where $N \ll HWZ$ and $N$ is determined by a predefined keeping ratio $R_{keep}$. After the pruning, all the following dense convolutions are replaced by corresponding sparse convolutions. Since sparse deconvolution will dilate the sparse features to empty positions and reduce the sparsity, we conduct similar pruning operations after each upsampling to maintain the spatial sparsity. Finally, we obtain a high-resolution and sparse occupancy feature $\mathbf{O}_{sparse} \in \mathbb{R}^{N' \times D'}$, where $N' \ll H'W'Z'$. Figure 2 illustrates the occupancy sparsify process.

## 4 EXPERIMENT

### 4.1 DATASETS

**nuScenes dataset** (Caesar et al., 2020) contains 1000 scenes in total, split into 700 in the training set, 150 in the validation set, and 150 in the test set. Each sequence is captured at 20Hz frequency with 20 seconds duration. Each sample contains RGB images from 6 cameras with 360° horizontal FOV and point cloud data from 32 beam LiDAR sensor. For the task of object detection, the key samples are annotated at 2Hz with ground truth labels for 10 foreground object classes (*thing*). For the task of semantic segmentation and panoptic segmentation, every point in the key samples is annotated using 6 more background classes (*stuff*) in addition to the 10 foreground classes (*thing*).

**Occ3D-nuScenes** (Tian et al., 2023) contains 700 training scenes and 150 validation scenes. The occupancy scope is defined as $-40m$ to $40m$ for X and Y-axis, and $-1m$ to $5.4m$ for the Z-axis in the ego coordinate. The voxel size is $0.4m \times 0.4m \times 0.4m$ for the occupancy label. The semantic labels contain 17 categories (including 'others'). Besides, it also provides visibility masks for LiDAR and camera modality, indicating which regions are visible from the sensor.

**Evaluation metrics.** nuScenes dataset uses mean Average Precision (mAP) and nuScenes Detection Score (NDS) metrics for the detection task, mean Intersection over Union (mIoU), and Panoptic Quality (PQ) metrics (Kirillov et al., 2019) for the semantic and panoptic segmentation. PQ$^{\dagger}$ is a modified panoptic quality (Porzi et al., 2019), which maintains the PQ metric for *thing* classes, but modifies the metric for *stuff* classes. The Occ3D-nuScenes benchmark calculates the mean Intersection over Union (mIoU) for 17 semantic categories within the camera's visible region.

### 4.2 EXPERIMENTAL SETTINGS

**Implementation Details.** For the implementation details of the model, please refer to the Section A.1 in the appendix. On the nuScenes dataset (Caesar et al., 2020), we set the point cloud range for the $x$ and $y$ axes to $[-51.2m, 51.2m]$, and $[-5m, 3m]$ for the $z$ axis. The voxel grid size used for loss supervision is $(0.256m, 0.256m, 0.125m)$. We trained the model on 8 NVIDIA A100 GPUs with a batch size of 1 per GPU. During training, we utilized the AdamW (Loshchilov & Hutter, 2017) optimizer for 24 epochs, with an initial learning rate of $3 \times 10^{-4}$ and the step schedule in $[20, 23]$. The input image size is cropped to $640 \times 1600$. When using the R101-DCN (Dai et al., 2017b) or InternImage (Wang et al., 2022a) as the backbone, we default to the 1.0 image scale. However, when using the R50 (He et al., 2016) backbone, we adopt a 0.5 image scale. More details please refer to the Section A.3 in the appendix.

**Evaluation.** For the sparse evaluation on the LiDAR benchmark, our approach can evaluate LiDAR semantic segmentation by assigning voxel semantic predictions to LiDAR points. We further extend it with object detection results, enabling panoptic evaluation on the LiDAR panoptic segmentation (Fong et al., 2022). As PQ is only computed on sparse points and cannot comprehensively reflect the understanding of foreground objects, we still choose to use mAP, NDS, and mIoU to measure the effectiveness of our approach in the experiments. For the dense evaluation on the occupancy benchmark, we directly compute the mIoU based on the occupancy label.

### 4.3 MAIN RESULTS

We validate our methods' performance on three benchmarks: 3D semantic segmentation, 3D panoptic segmentation and 3D occupancy prediction within the nuScenes dataset. The results demonstrate that our PanoOcc achieves state-of-the-art performance across all benchmarks. Notably, we are also the first to implement an end-to-end method for camera-based panoptic segmentation.

**3D Semantic Segmentation.** As shown in Table 1 and Table 2, we evaluate the semantic segmentation performance on the nuScenes test and validation set. In Table 1, we adopt the ResNet101-DCN (Dai et al., 2017b) initialized from FCOS3D (Wang et al., 2021) checkpoint, the same setting as (Huang et al., 2023) and (Zhang et al., 2023). Without bells and whistles, our PanoOcc surpass all the previous camera-based methods. In Table 2, we adopt three types of backbone to conduct experiments. Under the R50 (He et al., 2016) and ResNet101-DCN (Dai et al., 2017b) setting, our method achieves 68.1 mIoU and 71.6 mIoU, a new state-of-the-art. To further validate our approach, we experiment with a larger image backbone (Wang et al., 2022a) and achieve an impressive 74.5 mIoU, approaching the performance of current state-of-the-art LiDAR-based methods.

| Method | Input Modality | mIoU | barrier | bicycle | bus | car | const. veh. | motorcycle | pedestrian | traffic cone | trailer | truck | drive. suf. | other flat | sidewalk | terrain | manmade | vegetation |
|---|---|---|---|---|---|---|---|---|---|---|---|---|---|---|---|---|---|---|
| MINet Li et al. (2021) | LiDAR | 56.3 | 54.6 | 8.2 | 62.1 | 76.6 | 23.0 | 58.7 | 37.6 | 34.9 | 61.5 | 46.9 | 93.3 | 56.4 | 63.8 | 64.8 | 79.3 | 78.3 |
| PolarNet Zhang et al. (2020) | LiDAR | 69.4 | 72.2 | 16.8 | 77.0 | 86.5 | 51.1 | 69.7 | 64.8 | 54.1 | 69.7 | 63.5 | 96.6 | 67.1 | 77.7 | 72.1 | 87.1 | 84.5 |
| PolarSteam Chen et al. (2021) | LiDAR | 73.4 | 71.4 | 27.8 | 78.1 | 82.0 | 61.3 | 77.8 | 75.1 | 72.4 | 79.6 | 63.7 | 96.0 | 66.5 | 76.9 | 73.0 | 88.5 | 84.8 |
| JS3C-Net Yan et al. (2021) | LiDAR | 73.6 | 80.1 | 26.2 | 87.8 | 84.5 | 55.2 | 72.6 | 71.3 | 66.3 | 76.8 | 71.2 | 96.8 | 64.5 | 76.9 | 74.1 | 87.5 | 86.1 |
| AMVNet Liong et al. (2020) | LiDAR | 77.3 | 80.6 | 32.0 | 81.7 | 88.9 | 67.1 | 84.3 | 76.1 | 73.5 | 84.9 | 67.3 | 97.5 | 67.4 | 79.4 | 75.5 | 91.5 | 88.7 |
| SPVNAS Tang et al. (2020) | LiDAR | 77.4 | 80.0 | 30.0 | 91.9 | 90.8 | 64.7 | 79.0 | 75.6 | 70.9 | 81.0 | 74.6 | 97.4 | 69.2 | 80.0 | 76.1 | 89.3 | 87.1 |
| Cylinder3D++ Zhu et al. (2021) | LiDAR | 77.9 | 82.8 | 33.9 | 84.3 | 89.4 | 69.6 | 79.4 | 77.3 | 73.4 | 84.6 | 69.4 | 97.7 | 70.2 | 80.3 | 75.5 | 90.4 | 87.6 |
| AF2S3Net Cheng et al. (2021) | LiDAR | 78.3 | 78.9 | 52.2 | 89.9 | 84.2 | 77.4 | 74.3 | 77.3 | 72.0 | 83.9 | 73.8 | 97.1 | 66.5 | 77.5 | 74.0 | 87.7 | 86.8 |
| DRINet++ Ye et al. (2021) | LiDAR | 80.4 | 85.5 | 43.2 | 90.5 | 92.1 | 64.7 | 86.0 | 83.0 | 73.3 | 83.9 | 75.8 | 97.0 | 71.0 | 81.0 | 77.7 | 91.6 | 90.2 |
| LidarMultiNet Ye et al. (2022) | LiDAR | 81.4 | 80.4 | 48.4 | 94.3 | 90.0 | 71.5 | 87.2 | 85.2 | 80.4 | 86.9 | 74.8 | 97.8 | 67.3 | 80.7 | 76.5 | 92.1 | 89.6 |
| TPVFormer Huang et al. (2023) | Camera | 69.4 | 74.0 | 27.5 | 86.3 | 85.5 | **60.7** | 68.0 | 62.1 | 49.1 | 81.9 | 68.4 | 94.1 | 59.5 | 66.5 | 63.5 | 83.8 | 79.9 |
| OccFormer Zhang et al. (2023) | Camera | 70.8 | 72.8 | 29.9 | 87.9 | **85.6** | 57.1 | 74.9 | 63.2 | 53.4 | **83.0** | 67.6 | 94.8 | 61.9 | 70.0 | 66.0 | **84.0** | **80.5** |
| PanoOcc(Ours) | Camera | 71.4 | **82.5** | **32.3** | **88.1** | 83.7 | 46.1 | **76.5** | **67.6** | **53.6** | 82.9 | **69.5** | **96.0** | 66.3 | **72.3** | **66.3** | 80.5 | 77.3 |

Table 1: **LiDAR semantic segmentation results on nuScenes test set.** Our method achieves new state-of-the-art performance on camera-based semantic segmentation. For a fair comparison, we use the same backbone R101-DCN and train for 24 epochs.

| Method | Input Modality | Image Backbone | mIoU | barrier | bicycle | bus | car | const. veh. | motorcycle | pedestrian | traffic cone | trailer | truck | drive. suf. | other flat | sidewalk | terrain | manmade | vegetation |
|---|---|---|---|---|---|---|---|---|---|---|---|---|---|---|---|---|---|---|---|
| RangeNet++ (Milioto et al., 2019) | LiDAR | - | 65.5 | 66.0 | 21.3 | 77.2 | 80.9 | 30.2 | 66.8 | 69.6 | 52.1 | 54.2 | 72.3 | 94.1 | 66.6 | 63.5 | 70.1 | 83.1 | 79.8 |
| PolarNet (Zhang et al., 2020) | LiDAR | - | 71.0 | 74.7 | 28.2 | 85.3 | 90.9 | 35.1 | 77.5 | 71.3 | 58.8 | 57.4 | 76.1 | 96.5 | 71.1 | 74.7 | 74.0 | 87.3 | 85.7 |
| Salsanext (Cortinhal et al., 2020) | LiDAR | - | 72.2 | 74.8 | 34.1 | 85.9 | 88.4 | 42.2 | 72.4 | 72.2 | 63.1 | 61.3 | 76.5 | 96.0 | 70.8 | 71.2 | 71.5 | 86.7 | 84.4 |
| Cylinder3D (Zhu et al., 2021) | LiDAR | - | 76.1 | 76.4 | 40.3 | 91.2 | 93.8 | 51.3 | 78.0 | 78.9 | 64.9 | 62.1 | 84.4 | 96.8 | 71.6 | 76.4 | 75.4 | 90.5 | 87.4 |
| RPVNet (Xu et al., 2021) | LiDAR | - | 77.6 | 78.2 | 43.4 | 92.7 | 93.2 | 49.0 | 85.7 | 80.5 | 66.0 | 66.9 | 84.0 | 96.9 | 73.5 | 75.9 | 76.0 | 90.6 | 88.9 |
| TPVFormer (Huang et al., 2023) | Camera | R50 | 59.3 | 64.9 | 27.0 | 83.0 | 82.8 | 38.3 | 27.4 | 44.9 | 24.0 | 55.4 | 73.6 | 91.7 | 60.7 | 59.8 | 61.1 | 78.2 | **76.5** |
| PanoOcc | Camera | R50 | 68.1 | 70.7 | 37.9 | 92.3 | 85.0 | 50.7 | 64.3 | 59.4 | 35.3 | 63.8 | 81.6 | 94.2 | 66.4 | 64.8 | 68.0 | 79.1 | 75.6 |
| BEVFormer (Li et al., 2022d) | Camera | R101-DCN | 56.2 | 54.0 | 22.8 | 76.7 | 74.0 | 45.8 | 53.1 | 44.5 | 24.7 | 54.7 | 65.5 | 88.5 | 58.1 | 50.5 | 52.8 | 71.0 | 63.0 |
| TPVFormer (Huang et al., 2023) | Camera | R101-DCN | 68.9 | 70.0 | 40.9 | 93.7 | 85.6 | 49.8 | **68.4** | 59.7 | 38.2 | 65.3 | 83.0 | 93.3 | 64.4 | 64.3 | 64.5 | 81.6 | 79.3 |
| OccFormer (Zhang et al., 2023) | Camera | R101-DCN | 70.4 | 70.3 | **43.8** | 93.2 | 85.2 | 52.0 | 59.1 | **67.6** | **45.4** | 64.4 | 84.5 | 93.8 | 68.2 | **67.8** | **68.3** | **82.1** | **80.4** |
| PanoOcc | Camera | R101-DCN | 71.6 | 74.3 | 43.7 | **95.4** | **87.0** | **56.1** | 64.6 | 66.2 | 41.4 | **71.5** | **85.9** | **95.1** | **70.1** | 67.0 | 68.1 | 80.9 | 77.4 |
| PanoOcc | Camera | Intern-XL | **74.5** | 75.3 | 51.1 | 96.9 | 87.5 | 56.6 | 85.6 | 68.0 | 43.0 | 74.1 | 87.1 | 95.1 | 71.0 | 68.7 | 70.3 | 82.3 | 79.3 |

Table 2: **LiDAR semantic segmentation results on nuScenes validation set.** Our method achieves comparable performance with state-of-the-art LiDAR-based methods and notably surpasses the recently proposed camera-based methods.

**3D Occupancy Prediction.** In Table 3, we evaluate our method for 3D occupancy prediction on the Occ3D-nuScenes validation set. All methods utilize camera input and are trained for 24 epochs. The performance of MonoScene (Cao & de Charette, 2022), BEVDet (Huang et al., 2021), BEV-Former (Li et al., 2022d), and CTF-Occ (Tian et al., 2023) is reported in the work of (Tian et al., 2023). The use of the camera visible mask during training has proven to be an effective technique. We re-implemented BEVFormer (Li et al., 2022d) with the inclusion of the camera mask during training. Similarly, BEVDet (Huang et al., 2021) also adopts this trick and reports improved performance on its official code repository. Our PanoOcc also use camera visibile mask during training and achieves a new state-of-art performance. We adopt the R101-DCN as the backbone and use 4 frames for temporal fusion.

| Method | Image Backbone | mIoU | others | barrier | bicycle | bus | car | const. veh. | motorcycle | pedestrian | traffic cone | trailer | truck | drive. suf. | other flat | sidewalk | terrain | manmade | vegetation |
|---|---|---|---|---|---|---|---|---|---|---|---|---|---|---|---|---|---|---|---|
| MonoScene (Cao & de Charette, 2022) | R101-DCN | 6.06 | 1.75 | 7.23 | 4.26 | 4.93 | 9.38 | 5.67 | 3.98 | 3.01 | 5.90 | 4.45 | 7.17 | 14.91 | 6.32 | 7.92 | 7.43 | 1.01 | 7.65 |
| BEVDet (Huang et al., 2021) | R101-DCN | 11.73 | 2.09 | 15.29 | 0.0 | 4.18 | 12.97 | 1.35 | 0.0 | 0.43 | 0.13 | 6.59 | 6.66 | 52.72 | 19.04 | 26.45 | 21.78 | 14.51 | 15.26 |
| BEVFormer (Li et al., 2022d) | R101-DCN | 26.88 | 5.85 | 37.83 | 17.87 | 40.44 | 42.43 | 7.36 | 23.88 | 21.81 | 20.98 | 22.38 | 30.70 | 55.35 | 28.36 | 36.0 | 28.06 | 20.04 | 17.69 |
| CTF-Occ (Tian et al., 2023) | R101-DCN | 28.53 | 8.09 | 39.33 | 20.56 | 38.29 | 42.24 | 16.93 | 24.52 | 22.72 | 21.05 | 22.98 | 31.11 | 53.33 | 33.84 | 37.98 | 33.23 | 20.79 | 18.0 |
| BEVFormer* (Li et al., 2022d) | R101-DCN | 39.24 | 10.13 | 47.91 | 24.9 | 47.57 | 54.52 | 20.23 | 28.85 | 28.02 | 25.73 | 33.03 | 38.56 | 81.98 | 40.65 | 50.93 | 53.02 | 43.86 | 37.15 |
| BEVDet† (Huang et al., 2021) | Swin-B | 42.02 | 12.15 | 49.63 | 25.10 | 52.02 | 54.46 | 27.87 | 27.99 | 28.94 | 27.23 | 36.43 | 42.22 | 82.31 | 43.29 | 54.62 | 57.9 | 48.61 | 43.55 |
| PanoOcc | R101-DCN | **42.13** | 11.67 | 50.48 | 29.64 | 49.44 | 55.52 | 23.29 | 33.26 | 30.55 | 30.99 | 34.43 | 42.57 | 83.31 | 44.23 | 54.40 | 56.04 | 45.94 | 40.40 |

Table 3: **3D Occupancy prediction performance on the Occ3D-nuScenes dataset.** † denotes the performance is reported by its official code implementation. * means the performance is achieved by our implementation using the camera mask during training.

**3D Panoptic Segmentation.** PanoOcc is the first work to implement an end-to-end train model for camera-based panoptic segmentation. We compare our method with previous LiDAR-based panoptic segmentation methods. The results in Table 4 show that our PanoOcc achieves 62.1 PQ, demonstrating comparable performance to some LiDAR-based methods such as EfficientLPS (Sirohi et al., 2021) and PolarNet (Zhang et al., 2020). However, our approach still has a performance gap compared to state-of-the-art LIDAR-based methods, which can be attributed to the inferior detection performance (48.4 mAP v.s. 63.8 mAP).

| Method | Input Modality | PQ | PQ† | RQ | SQ | mAP |
|---|---|---|---|---|---|---|
| EfficientLPS Sirohi et al. (2021) | LiDAR | 62.0 | 65.6 | 73.9 | 83.4 | / |
| Panoptic-PolarNet Zhou et al. (2021) | LiDAR | 63.4 | 67.2 | 75.3 | 83.9 | / |
| Panoptic-PHNet Li et al. (2022a) | LiDAR | 74.7 | 77.7 | 84.2 | 88.2 | / |
| LidarMulitiNet Ye et al. (2022) | LiDAR | 81.8 | / | 90.8 | 89.7 | 63.8 |
| PanoOcc | Camera | 62.1 | 66.2 | 75.1 | 82.1 | 48.4 |

Table 4: **LiDAR panoptic segmentation results on nuScenes validation set.** Our PanoOcc based on the camera input has approached LiDAR-based methods' performance.

## 4.4 ABLATION

In this section, we mainly validate the key design choices of PanoOcc on the nuScenes validation set. More ablation studies of the model design please refer to Section A.2 in the appendix.

**Effectiveness of Joint Detection and Segmentation.** Table 5 demonstrates the significant positive impact of training for joint detection and segmentation. When compared to single-task models, the jointly-trained model excels in both the segmentation and detection tasks. Voxel selection further enhances the interaction between detection and segmentation learning, improving performance in both tasks. The unified occupancy representation also enables efficient learning of voxel features.

| | Det. | Seg. | Vox. Sel. | mIoU | mAP | NDS |
|---|---|---|---|---|---|---|
| (a) | ✓ | | | / | 0.252 | 0.310 |
| (b) | | ✓ | | 0.652 | / | / |
| (c) | ✓ | ✓ | | 0.656 | 0.266 | 0.319 |
| (d) | ✓ | ✓ | ✓ | **0.661** | **0.271** | **0.324** |

Table 5: **Effectiveness of joint detection and segmentation.** Det. stands for detection head. Seg. denotes segmentation head. Vox. Sel. represents voxel selection for foreground voxels.

**Efficiency of Coarse-to-Fine Design.** Table 6 illustrates the advantages of our coarse-to-fine scheme, which utilizes a low-resolution 3D voxel grid. This approach not only helps in increasing performance and inference speed but also effectively reduces memory consumption. By comparing it with the direct use of high-resolution voxel queries (200x200x8), we observe that our coarse-to-fine design achieves comparable or even superior performance while consuming nearly half the memory. This showcases the efficiency and effectiveness of our design choice.

| Voxel Resolution | Voxel Upsampling | Memory | Latency | Param | FPS | mIoU |
|---|---|---|---|---|---|---|
| 200x200x8 | | 37G / 9.5G | 255 ms | 117.7 M | 4.1 | 67.9 |
| 50x50x16 | ✓ | **18G / 5.7G** | **149 ms** | **48.7 M** | **9.2** | **68.3** |

Table 6: **Ablation study for the coarse-to-fine design.** We show the train / inference memory consumption, respectively. The experiments were conducted on the A100 GPU.

## 4.5 DISCUSSION

| Method | Query form | Resolution | Memory | Latency | FPS | mIoU |
|---|---|---|---|---|---|---|
| TPVFormer* | 2D Tri-plane | 200x(200+16+16) | 33.5G / 7.1G | 268 ms | 3.7 | 68.9 |
| PanoOcc | 3D Voxel | 50x50x16 | **24G / 6.0G** | **203 ms** | **4.8** | **71.6** |

Table 7: **Model efficiency comparison with different query forms**. The symbol * denotes performance obtained using the official code and released checkpoints. We report the train / inference memory consumption in the experiment.

**Voxel v.s. Tri-plane.** Traditionally, it has been widely believed that using 3D voxel grids alone is an inefficient solution due to the memory cost. This has led methods like TPVFormer (Huang et al., 2023) to split the 3D representation into three 2D planes. However, we have demonstrated for the first time that using the coarse-to-fine voxel representation can solve the memory increasing problem. In Table 7, we compare the performance and efficiency of our method with the previous state-of-the-art approach, TPVFormer (Huang et al., 2023), under the same experimental setup. Despite having an additional detection branch and the capability to output detection results, our model still exhibits lower memory consumption and faster inference speed.

**Occupancy Sparsify.** In contrast to 2D space, 3D space exhibits high sparsity, indicating that the majority of voxels are empty. In Table 8, we investigate the effectiveness of the occupancy sparsify strategy. Here we have 3 layers of sparse deconvolution for upsampling in total. In coarse-to-fine order, the keeping ratio after each upsampling is 0.2, 0.5, and 0.5, respectively. It suggests that finally we only keep 5% voxels, and this reduction has not resulted in a significant performance decrease.

| | Convolution | Latency | Memory | FPS | mIoU |
|---|---|---|---|---|---|
| (a) | Dense | 126 ms | 15 G | 9.3 | **0.654** |
| (b) | Sparse | **112 ms** | **9 G** | **9.7** | 0.639 |

Table 8: **Exploration of sparse architecture design.** The experiment is conducted under the R50 setting without temporal fusion.

**Temporal Enhancement.** In Table 9, we compared the impact of temporal information on different categories. The findings revealed that the semantic segmentation performance improved for almost all categories except for the barrier category. The motorcycle and trailer categories demonstrated a significant improvement, with a boost of 11.7 mIoU and 8.2 mIoU, respectively. These two categories are typically affected by occlusion, and thus, the utilization of temporal information can enhance the model's ability to accurately detect and segment occluded objects.

| mIoU | barrier | bicycle | bus | car | const. veh. | motorcycle | pedestrian | traffic cone | trailer | truck | drive. suf. | other flat | sidewalk | terrain | manmade | vegetation |
|---|---|---|---|---|---|---|---|---|---|---|---|---|---|---|---|---|
| 65.6 | 72.3 | 35.8 | 91.4 | 84.4 | 47.2 | 52.6 | 57.7 | 31.5 | 55.6 | 80.6 | 94.0 | 64.3 | 63.2 | 66.5 | 77.7 | 73.9 |
| 68.1 | 70.7 | 37.9 | 92.3 | 85.0 | 50.7 | 64.3 | 59.4 | 35.3 | 63.8 | 81.6 | 94.2 | 66.4 | 64.8 | 68.0 | 79.1 | 75.6 |
| (2.5↑) | (1.6↓) | (2.1↑) | (0.9↑) | (0.6↑) | (3.5↑) | (11.7↑) | (1.7↑) | (3.8↑) | (8.2↑) | (1.0↑) | (0.2↑) | (2.1↑) | (1.6↑) | (1.5↑) | (1.4↑) | (1.7↑) |

Table 9: **Effect of temporal enhancement on different categories.** The findings indicated that incorporating temporal information improved segmentation performance for most categories.

## 5 CONCLUSION

In this paper, we propose *camera-based 3D panoptic segmentation*, aiming for a comprehensive understanding of the scene by a unified occupancy representation. To facilitate occupancy representation learning, we propose a novel framework called PanoOcc that utilizes voxel queries to incorporate information from multi-frame and multi-view images in a coarse-to-fine scheme. Extensive experiments on the nuScenes dataset and Occ3D-nuScenes demonstrate the effectiveness of PanoOcc and its potential to advance holistic 3D scene understanding. We envision 3D occupancy representation as a promising new paradigm for future 3D scene perception.

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

# A APPENDIX

## A.1 IMPLEMENTATION DETAILS OF PANOOCC

In this section, we introduce the implementation details of PanoOcc.

**Image Backbone.** The backbone used in our approach includes ResNet50 (He et al., 2016), ResNet101-DCN (Dai et al., 2017b), and InternImage-XL (Wang et al., 2022a), with output multi-scale features from FPN (Lin et al., 2017a) at sizes of 1/8,1/16,1/32 and 1/64.

**Voxel Queries.** The initial resolution of the voxel queries is 50x50x16 for $H, W, Z$. We use an embedding dimension $D$ of 256, and learnable 3D position encoding is added to the voxel queries.

**Occupancy Encoder.** The camera view encoder includes 3 layers, with each layer consisting of voxel self-attention, voxel cross-attention, norm layer, and feed-forward layer, with both $M_1$ and $M_2$ set to 4. The temporal encoder fuses 4 frames (including the current frame) with a time interval of 0.5s. Our key difference from previous BEV-based methods primarily lies in the learning of voxel features. We designed voxel cross-attention and voxel self-attention to facilitate the interaction between multi-scale image features and voxel queries.

- **Voxel Cross-Attention**: Specifically, for a voxel query $\mathbf{q}$ located at $(i, j, k)$, the process of voxel cross-attention (VCA) can be formulated as follows:

$$\text{VCA}(\mathbf{q}, \mathbf{F}) = \frac{1}{|v|} \sum_{n \in v} \sum_{m=1}^{M_1} \text{DA}(\mathbf{q}, \pi_\mathbf{n}(\mathbf{Ref}_{i,j,k}^m), \mathbf{F}_n) \tag{4}$$

  where $n$ indexes the camera view, $m$ indexes the reference points, and $M_1$ is the total number of sampling points for each voxel query. $v$ is the set of image views for which the projected 2D point of the voxel query can fall on. $\mathbf{F}_n$ is the image features of the $n$-th camera view. $\pi_\mathbf{n}(\mathbf{Ref}_{i,j,k}^m)$ denotes the $m$-th projected reference point in $n$-th camera view, projected by projection matrix $\pi_\mathbf{n}$ from the voxel grid located at $(i, j, k)$. DA represents deformable attention. The real position of a reference point located at voxel grid $(i, j, k)$ in the ego-vehicle frame is $(x_i^m, y_j^m, z_k^m)$. The projection between $m$-th projected reference point $\mathbf{Ref}_{i,j,k}^m$ and its corresponding 2D reference point $(u_{ijk}^{n,m}, v_{ijk}^{n,m})$ on the $n$-th view can be formulate as:

$$\mathbf{Ref}_{i,j,k}^m = (x_i^m, y_j^m, z_k^m) \tag{5}$$

$$d_{ijk}^{n,m} \cdot [u_{ijk}^{n,m}, v_{ijk}^{n,m}, 1] = \mathbf{P}_\mathbf{n} \cdot [x_i^m, y_j^m, z_k^m, 1]^T \tag{6}$$

  where $\mathbf{P}_n \in \mathbb{R}^{3 \times 4}$ is the projection matrix of the $n$-th camera. $(u_{ijk}^{n,m}, v_{ijk}^{n,m})$ denotes the $m$-th 2D reference point on $n$-th image view. $d_{ijk}^{n,m}$ is the depth in the camera frame.

- **Voxel Self-Attention**: Voxel self-attention (VSA) facilitates the interaction between voxel queries. For a voxel query $\mathbf{q}$ located at $(i, j, k)$, it only interacts with the voxel queries at the reference points nearby. The process of voxel self-attention can be formulated as follows:

$$\text{VSA}(\mathbf{q}, \mathbf{Q}) = \sum_{m=1}^{M_2} \text{DA}(\mathbf{q}, \mathbf{Ref}_{i,j,k}^m, \mathbf{Q}) \tag{7}$$

  where $m$ indexes the reference points, and $M_2$ is the total number of reference points for each voxel query. DA represents deformable attention. Contrary to the reference points on the image plane in voxel cross-attention, $\mathbf{Ref}_{i,j,k}^m$ in voxel self-attention is defined on the BEV plane.

$$\mathbf{Ref}_{i,j,k}^m = (x_i^m, y_j^m, z_k) \tag{8}$$

  where $(x_i^m, y_j^m, z_k)$ denotes the $m$-th reference point for query $\mathbf{q}$. These sampling points share the same height $z_k$, but with different learnable offsets for $(x_i^m, y_j^m)$. This encourages the voxel queries to interact in the BEV plane, which contains more semantic information.

**Occupancy Decoder.** The voxel upsample module employs 3 layers of 3D deconvolutions to up-scale 4x for $H$ and $W$, and 2x for $Z$, with detailed parameters in the Table 10. The upsampled voxel features have dimensions of 200x200x32 for $H', W', Z'$, and a feature dimension $D'$ of 64.

| Hyperparameters | Values |
|---|---|
| #Input features | 50x50x16x256 (H,W,Z,D) |
| #Output features | 200x200x32x64 (H',W',Z',D') |
| ConvTranspose3D#1 | kernel:(1,5,5), stride:(1,1,1) |
| ConvTranspose3D#2 | kernel:(1,4,4), stride:(1,2,2) |
| ConvTranspose3D#3 | kernel:(2,4,4), stride:(2,2,2) |
| Activate function | ReLU |
| Normalize | BN3D |

Table 10: **Network hyper-parameters of voxel upsample module.**

**Task Head.** The segmentation head has 2 MLP layers with a hidden dimension of 128 and uses *soft-plus* (Zheng et al., 2015) as the activation function. The number of object queries for the detection head is set to 900, and has 6 layers decoder, similar to (Li et al., 2022d).

## A.2   ABLATION STUDIES ON MODEL DESIGN

**Initial Voxel Resolution.** Table 11 compares the results of different initial resolutions used for voxel queries in our experiments. In experiments (b), (c), and (d), we maintained fixed dimensions of $H$ and $W$ while varying the resolution of $Z$. Our findings clearly demonstrate that encoding height information is a crucial factor in achieving superior performance in both segmentation(+5.3 mIoU) and detection tasks(+1.2 mAP and +1.6 NDS), with a more significant impact observed in segmentation tasks. Furthermore, we observed that (a) and (b) have the same number of query parameters and perform similarly in detection tasks. However, there is a significant gap in the segmentation tasks between these two. Specifically, the mIoU gain from (d) to (a) is much less compared to that from (d) to (b). The experiment (e) results suggest that when the dimensions of $H$ and $W$ are too small, there will be a significant reduction in the performance of both detection and segmentation tasks. Overall, our findings emphasize the importance of encoding height information to achieve fine-grained scene understanding.

| | Query Resolution | mIoU | mAP | NDS |
|---|---|---|---|---|
| (a) | 100x100x4 | 0.617 | **0.276** | **0.327** |
| (b) | 50x50x16 | **0.661** | 0.271 | 0.324 |
| (c) | 50x50x8 | 0.631 | 0.267 | 0.316 |
| (d) | 50x50x4 | 0.608 | 0.259 | 0.308 |
| (e) | 25x25x16 | 0.591 | 0.244 | 0.294 |

Table 11: **Ablation study for different initial query resolutions.** Height information is important to achieve fine-grained 3D scene understanding.

**Design of Camera View Encoder.** Table 12 presents the ablation study conducted on the design choices in the camera view encoder. Specifically, we experimented with different combinations of attention modules in (b), (c), and (d). The results demonstrated that incorporating voxel self-attention (VSA) enhanced the interaction between queries, leading to improved performance. Considering both performance and parameters, we choose 3 layers as default.

| | Layers | Attention module | mIoU | mAP | NDS |
|---|---|---|---|---|---|
| (a) | 1 | VSA + VCA | 0.648 | 0.251 | 0.294 |
| (b) | 3 | VCA | 0.644 | 0.264 | 0.312 |
| (c) | 3 | VSA + VCA | 0.653 | 0.267 | 0.314 |
| (d) | 3 | VSA×2 + VCA | 0.661 | **0.271** | **0.324** |
| (e) | 6 | VSA×2 + VCA | **0.662** | 0.267 | 0.319 |

Table 12: **Ablation study for camera view encoder.** VSA denotes voxel self-attention, while VCA means voxel cross-attention.

**Design of Temporal Encoder.** Table 13 presents extensive ablation studies on the design of the temporal encoder, including different time intervals, number of frames, fusion methods, and encoder network architectures. Compared to (a) and (b) designs, both detection and segmentation tasks show a significant improvement (+2.5 mIoU, +2.4 mAP, and +7.1 NDS), which suggests the importance of temporal information. In (b)(c)(d), we compared the influence of different time intervals and found that longer intervals do not improve the fine-grained segmentation performance. In (e) and (f), we also compared different ways to fuse the historical features and found that directly concatenating the features performs better than using temporal self-attention (Li et al., 2022d).

| | Temp. | Intv. | Frames | Fuse | Arch. | mIoU | mAP | NDS |
|---|---|---|---|---|---|---|---|---|
| (a) | | / | 1 | / | C3D×1 | 0.656 | 0.269 | 0.319 |
| (b) | ✓ | 0.5s | 4 | Cat. | C3D×1 | **0.681** | 0.293 | **0.390** |
| (c) | ✓ | 1s | 4 | Cat. | C3D×1 | 0.657 | **0.294** | 0.385 |
| (d) | ✓ | 2s | 4 | Cat. | C3D×1 | 0.660 | 0.294 | 0.375 |
| (e) | ✓ | 1s | 4 | Cat. | C3D×3 | 0.658 | 0.290 | 0.379 |
| (f) | ✓ | 0.5 | 4 | TSA | DA | 0.648 | 0.271 | 0.323 |

Table 13: **Ablation study for temporal encoder.** Temp. stands for temporal fusion, while ✓ denotes using temporal fusion. Intv. denotes time interval. Arch. refers to the architecture used in temporal encoder. C3D represents 3D convolution. ×3 means using 3 blocks of the architecture. Cat. means concatenating features from different frames, and TSA represents the temporal self-attention structure in (Li et al., 2022d). DA means deformable attention (Zhu et al., 2020).

**The Supervision for Voxel Representation.** Table 14 ablates the effects of different resolutions for segmentation loss supervision. The experiment results indicate that resolution at 400x400x64 has the best performance.

| Supervision | Voxel feats | Loss Resolution | mIoU | mAP | NDS |
|---|---|---|---|---|---|
| LiDAR | 200x200x32 | 400x400x64 | **0.661** | **0.271** | **0.324** |
| LiDAR | 200x200x32 | 200x200x32 | 0.644 | 0.267 | 0.316 |
| LiDAR | 100x100x16 | 100x100x16 | 0.609 | 0.264 | 0.317 |

Table 14: **Supervision for voxel representation.** We utilize sparse LiDAR point labels as the supervision for voxel representation.

**Loss Terms and Weights.** Table 15 presents the comparison of various combinations of loss terms and weights. It indicates that the $\mathcal{L}_{lovasz}$ plays a crucial role in the segmentation learning process, as its removal led to a significant drop in performance (from 65.6 to 59.6 mIoU). We also experimented with various weight combinations and found that $\lambda_1 = 10, \lambda_2 = 10, \lambda_3 = 5$ performs best.

## A.3 TRAINING AND INFERENCE DETAILS

**Training.** We trained the model on 8 NVIDIA A100 GPUs with a batch size of 1 per GPU. During training, we utilized the AdamW (Loshchilov & Hutter, 2017) optimizer for 24 epochs, with an initial learning rate of $3 \times 10^{-4}$ and the step schedule in [20, 23]. Additionally, we employed several data augmentation techniques, including image scaling, color distortion, and Gridmask (Chen et al., 2020). The input image size is cropped to $640 \times 1600$. The loss weights used in our approach are $\lambda_1$=10.0, $\lambda_2$=10.0, $\lambda_3$=5.0, $\lambda_4$=2.0, and $\lambda_5$=0.25.

**Supervision.** For the detection head, we use object-level annotations as the supervision. We employ sparse LiDAR point-level semantic labels for the segmentation head to supervise voxel prediction. When multiple semantic labels are present within a voxel grid, we prioritize the category label with the highest count of LiDAR points. As for the occupancy prediction, we rely on the occupancy label as the source of supervision.

| $\mathcal{L}_{focal}$ | $\mathcal{L}_{lovasz}$ | $\mathcal{L}_{thing}$ | $\lambda_1$ | $\lambda_2$ | $\lambda_3$ | mIoU | mAP | NDS |
|:---:|:---:|:---:|:---:|:---:|:---:|:---:|:---:|:---:|
| ✓ | | | 10.0 | / | / | 0.596 | 0.259 | 0.315 |
| ✓ | ✓ | | 10.0 | 10.0 | / | 0.656 | 0.266 | 0.319 |
| | ✓ | ✓ | / | 10.0 | 5.0 | 0.643 | 0.260 | 0.311 |
| ✓ | ✓ | ✓ | 10.0 | 10.0 | 5.0 | **0.661** | **0.271** | **0.324** |
| ✓ | ✓ | ✓ | 10.0 | 10.0 | 10.0 | 0.652 | 0.265 | 0.317 |
| ✓ | ✓ | ✓ | 5.0 | 10.0 | 5.0 | 0.656 | 0.266 | 0.315 |
| ✓ | ✓ | ✓ | 15.0 | 10.0 | 5.0 | 0.650 | 0.265 | 0.314 |
| ✓ | ✓ | ✓ | 10.0 | 15.0 | 5.0 | 0.654 | 0.263 | 0.312 |

Table 15: **Ablation for loss terms and weights.** We ablates different loss combinations and its weight.

### A.4 VISUALIZATION

Figure 3 showcases qualitative results achieved by PanoOcc on the nuScenes validation set. The voxel predictions are visualized at a resolution of 200x200x32 and assign to LiDAR points. These visualizations highlight the accuracy and reliability of our predictions for 3D semantic segmentation and panoptic segmentation. Figure 4 illustrates the dense occupancy prediction on the Occ3D-nuScenes validation set, where voxel predictions are visualized at the resolution of 200x200x16.

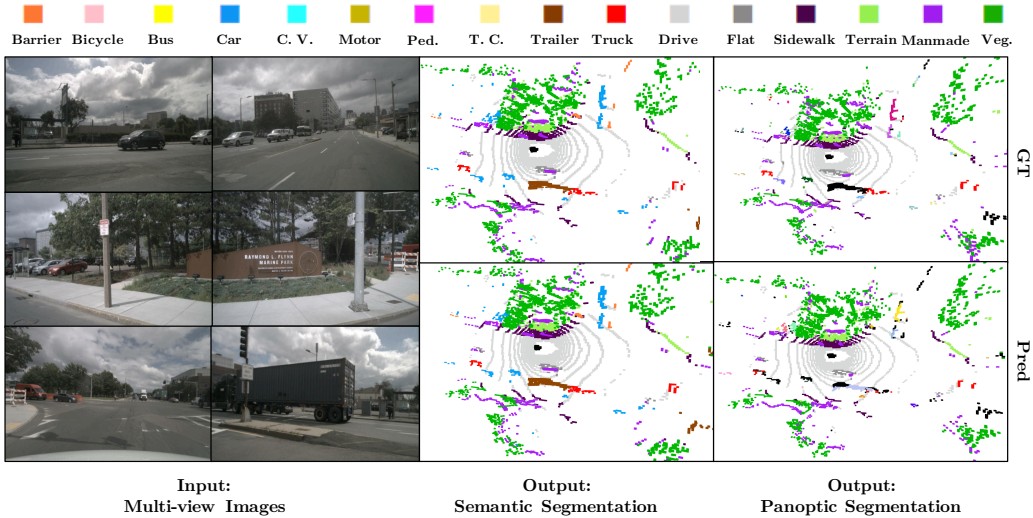

Figure 3: **Qualitative results on nuScenes validation set.** Our PanoOcc takes multi-view images as input and produces voxel predictions, which are visualized at a resolution of 200x200x32. We evaluate 3D segmentic segmentation and panoptic segmentation on LiDAR points.

## B REPRODUCIBILITY STATEMENTS

We are committed to providing the research community with the necessary resources to replicate our work. We will release the training and inference codes, accompanied by well-documented instructions to facilitate the replication process. Our codebase is built upon mmdetection3D[1], ensuring that it is user-friendly and accessible to the wider community. The data and annotations of nuScenes[2] are publicly available.

---

[1] https://github.com/open-mmlab/mmdetection3d
[2] https://nuscenes.org

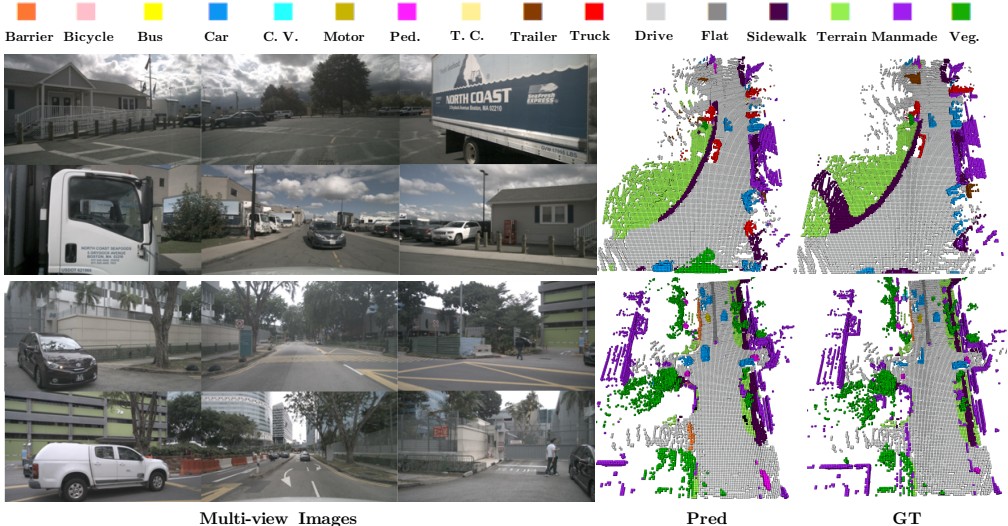

Figure 4: **Qualitative results on Occ3D-nuScenes validation set.** Our PanoOcc takes multi-view images as input and produces dense occupancy predictions, which are visualized at the resolution of 200x200x16.

