# OpenReview forum: "PanoOcc: Unified Occupancy Representation for Camera-based 3D Panoptic Segmentation"
_ICLR.cc/2024/Conference — ICLR 2024 Conference Withdrawn Submission_

### Official Review · Reviewer_eE1X · 2023-10-22

**Soundness:** 3 good
**Presentation:** 2 fair
**Contribution:** 2 fair
**Rating:** 5
**Confidence:** 3

**Summary:**

The paper introduces the PanoOcc method for camera-based 3D panoptic segmentation in autonomous driving scenarios. The proposed approach addresses the limitation of existing perception tasks by providing a unified occupancy representation for comprehensive 3D scene understanding. PanoOcc utilizes voxel queries to aggregate spatiotemporal information from multi-frame and multi-view images, integrating feature learning and scene representation. The method achieves state-of-the-art results for camera-based semantic segmentation and panoptic segmentation on the nuScenes dataset, and demonstrates promising performance on the Occ3D benchmark.

**Strengths:**

1. Comprehensive approach: The paper addresses the limitation of existing perception tasks by proposing a unified solution for camera-based 3D panoptic segmentation, incorporating object detection, semantic segmentation, and occupancy prediction into a single framework.
2. Efficient and effective: The PanoOcc method achieves state-of-the-art results while being computationally efficient, utilizing voxel queries and coarse-to-fine upsampling schemes to improve performance and reduce memory consumption.
3. Integration of spatiotemporal information: By incorporating spatiotemporal information from multi-frame and multi-view images, PanoOcc enhances the model's ability to detect and segment objects in complex scenes, especially occluded objects.

**Weaknesses:**

1. Lack of novelty: The paper's proposed method lacks novelty as it heavily relies on existing BEV detection models and sparse operations commonly used in 3D point cloud detection. It lacks innovation in terms of addressing the specific challenges and design considerations for occupancy tasks.
2. Limited generalization: The experiments are solely conducted on the nuScenes dataset, which may limit the generalizability of the proposed method to other autonomous driving scenarios or datasets. More extensive evaluation on diverse datasets is necessary to demonstrate the method's ability to generalize.
3. Limited performance improvement: Despite achieving state-of-the-art results, the paper reports only modest improvements in terms of mIoU. The performance on irregular objects is particularly lacking, which contradicts the initial intention of addressing occupancy tasks for handling irregular shapes.
4. Lack of in-depth analysis: The paper could benefit from providing more in-depth analysis and discussion on the limitations and potential failure cases of the proposed method. Further insights into the strengths and weaknesses of the approach would enhance the understanding of its limitations and avenues for future improvements.

**Questions:**

What I am primarily concerned about is that this article has limited novelty, as it stems from some extensions to existing BEV 3D detection techniques. However, it lacks specific design considerations for how to approach the occupancy task.

---

### Official Review · Reviewer_bsGt · 2023-10-29

**Soundness:** 3 good
**Presentation:** 3 good
**Contribution:** 2 fair
**Rating:** 5
**Confidence:** 5

**Summary:**

This paper designs a model that can jointly handle 3D objection and panoptic occupancy prediction tasks. The majority of the previous occupancy prediction methods can be seen as 3D semantic segmentation tasks. In this paper, With the help of the detection head, the model can identify different things. This paper shows that a joint learning paradigm can benefit both 3D detection and 3D segmentation.

**Strengths:**

1. The writing and figures of this paper are very clear. In fact, based on Figure 1, the reviewer can clearly understand the entire.
2. The adoption of sparse convolution solves the problem of excessive calculation of 3D voxels.

**Weaknesses:**

The contribution of this paper is trivial for the following reasons:

1. There is nothing new in the generation of occupation features. It basically refers to the existing BEV generation method. A similar approach has been used in Voxformer.
2. The joint detection and segmentation learning is too trivial. It is just like adding another detection task on top of an occupancy model. Using detection boxes to identify different instances from the occupancy results is also one common approach in the panoptic segmentation field.

**Questions:**

1. Although the paper claims to be a joint task of detection and segmentation. However, the detection results were not widely reported in the paper. The detection results in table 5 are only 31-32 NDS. The reviewer wonders the reason.

**Details Of Ethics Concerns:**

/

---

### Official Review · Reviewer_iHeu · 2023-10-30

**Soundness:** 3 good
**Presentation:** 3 good
**Contribution:** 1 poor
**Rating:** 3
**Confidence:** 4

**Summary:**

This paper introduces PanoOcc, which leverages voxel queries to aggregate spatiotemporal information from multi-frame and multi-view images in a coarse-to-fine scheme. The authors have validated the effectiveness and efficiency of PanoOcc through ablation studies, and claim to achieve satisfactory results for camera-based semantic and panoptic segmentation on the nuScenes dataset.

**Strengths:**

The target problem is important, the paper is well-written, and the experimental results are comprehensive.

**Weaknesses:**

A primary concern regarding this paper pertains to its level of technical novelty. While the methodologies employed, including voxel queries, temporal fusion, object detection, semantic segmentation, sparse convolution, and a coarse-to-fine structure are well-established in prior works, this paper appears to combine these existing techniques to attain camera-based panoptic occupancy prediction.

**Questions:**

I find myself inquiring about the core novelty of this paper. At a glance, it appears to be a straightforward combination of previously established ideas. Unfortunately, I didn’t learn any fresh insights or innovative contributions from this work.

---

### Official Review · Reviewer_DemZ · 2023-10-31

**Soundness:** 3 good
**Presentation:** 2 fair
**Contribution:** 3 good
**Rating:** 5
**Confidence:** 3

**Summary:**

The paper targets to solve the 3D panoptic segmentation problem on outdoor scenes. To solve this problem, the authors propose PanoOCC, a novel framework to do 3D panoptic segmentation. The proposed architecture leverages both view encoders and temporal encoders, and the output uses coarse-to-fine manner to predict panoptic segmentation including 3D detection and 3D segmentation. The proposed method is evaluated on nuscenes and OCC3D, achieving state-of-the-art results.

**Strengths:**

1. Coarse-to-fine and refinement on 3D techniques are not new, however it makes sense to validate their usage/effectiveness in the new task.
2. Thorough ablation studies are conducted on proposed modules.
3. The proposed method is evaluated on multiple major benchmarks.

**Weaknesses:**

1. Marginal improvements on most of the benchmarks.

2. Missing some details, such as how do you exactly aggregate the voxel features from multi-view images. What the projection matrix looks like, do you know the intrinsic matrix of color images etc.

3. Any ablation to show advantages of using a query-based method for this task?

**Questions:**

1. It is unclear to me how you would gather the multi-view images features for each voxel? Do you use depth or projecting the duplicated features along the ray?

2. How do you get the 3D instance segmentation? Instance-level 3D segment mask is also a part of panoptic segmentation evaluation.

In general, there are too many missing pieces, which leads to a hard time to understand the technical details.